# A Data Fusion Method for Non-Destructive Testing by Means of Artificial Neural Networks

**DOI:** 10.3390/s21082598

**Published:** 2021-04-07

**Authors:** Romain Cormerais, Aroune Duclos, Guillaume Wasselynck, Gérard Berthiau, Roberto Longo

**Affiliations:** 1ESEO-GSII, 10 Blvd Jean Jeanneteau, 49000 Angers, France; roberto.longo@eseo.fr; 2IREENA, 37 Blvd de l’Université, 44600 Saint-Nazaire, France; Guillaume.Wasselynck@univ-nantes.fr (G.W.); gerard.berthiau@univ-nantes.fr (G.B.); 3LAUM UMR CNRS 6613, Université du Mans, Av Olivier Messiaen, 72000 Le Mans, France; aroune.duclos@univ-lemans.fr

**Keywords:** Non Destructive Testing (NDT), Ultrasonic (US), Eddy Current (EC), Machine Learning (ML), Artificial Neural Networks (ANNs), data fusion

## Abstract

In the aeronautics sector, aircraft parts are inspected during manufacture, assembly and service, to detect defects eventually present. Defects can be of different types, sizes and orientations, appearing in materials presenting a complex structure. Among the different inspection techniques, Non Destructive Testing (NDT) presents several advantages as they are noninvasive and cost effective. Within the NDT methods, Ultrasonic (US) waves are widely used to detect and characterize defects. However, due the so-called blind zone, they cannot be easily employed for defects close to the surface being inspected. On the other hand, another NDT technique such Eddy Current (EC) can be used only for detecting flaws close to the surface, due to the presence of the EC skin effect. The work presented in this article aims to combine the use of these two NDT methods, exploiting their complementary advantages. To reach this goal, a data fusion method is developed, by using Machine Learning techniques such as Artificial Neural Networks (ANNs). A simulated training database involving simulations of US and EC signals propagating in an Aluminum block in the presence of Side Drill Holes (SDHs) has been implemented, to train the ANNs. Measurements have been then performed on an Aluminum block, presenting tree different SDHs at specific depths. The trained ANNs were used to characterize the different real SDHs, providing an experimental validation. Eventually, particular attention has been addressed to the estimation errors corresponding to each flaw. Experimental results will show that depths and radii estimations error were confined on average within a range of 4%, recording a peak of 11% for the second SDHs.

## 1. Introduction

Non Destructive Testing (NDT) is a wide group of analysis techniques employed to assess the quality and to evaluate the safety of materials in building or transport industry [1]. These noninvasive methods allow us to monitor materials during their manufacturing, assembling and their lifetime. The flaws to be detected can be of different types, sizes and orientations and may be in a wide variety of materials. Various techniques are used in NDT such as Ultrasonic (US), Eddy Current (EC), radiography, penetrating testing and thermography [2,3]. These methods present different advantages on their capabilities to detect flaws of specific shapes, on their implementations and on the nature of the materials that can be tested. Most NDT techniques make use of US to determine position and size of structural flaws in a wide variety of materials [4,5,6,7]. However, this technique presents a blind zone for the acoustic field, located near the sensor, making impossible the detection of surface flaws, if no specific coupling techniques are used [8]. Another well-known NDT technique is EC testing [9,10,11]. This method is used to detect flaws close to the inspected surface as opposed to US. However, subsurface flaws cannot be detected due to electromagnetic skin effect of EC [12]. As these two NDT techniques offer complementary advantages and disadvantages, their joint implementation could allow us to achieve more accurate detection and characterization of flaws. In addition, one could exploit redundant information eventually present in both US and EC signals.

This data analysis approach, making use of different sources treated simultaneously, is commonly called “data fusion” and is employed in a wide variety of fields [13,14,15], including NDT [16,17,18,19]. Some works already offered data fusion techniques for NDT, mainly relying on probabilistic approaches [20], using the fuzzy set theory [21] and the Dempster-Shafer theory [22]. The main drawbacks of these approaches consist of *a priori* knowledge and their difficulty to be automated. On the other hand, Machine Learning (ML) algorithms are another data fusion approach that overcomes these issues by learning directly from data [23,24,25]. Although ML algorithms have been used extensively to detect or characterize defects using EC [26,27,28] or US measurements [29,30,31], to our knowledge there are no NDT works dealing with simultaneous processing of EC and US signals.

Hence, the aim of this article is to propose a data fusion algorithm based on Artificial Neural Networks (ANNs), exploiting the simplicity and versatility of this particular ML algorithm and combing the advantages of both US and EC measurements. The ultimate goal of this study could lead to the design of an NDT smart sensor, where no need for particular coupling or *a priori* knowledge is required to get accurate estimations of flaws eventually present in the inspected structure.

The article is organized in two main sections. The section named Method presents the tools employed to build a simulated training data base and an experimental validation data base, as well as the ANNs architecture implementing the data fusion. Section 3 sums up the main data fusion achievements exploiting the combined use of EC and US measurements. Eventually, conclusions and perspective will be discussed at the end of the article.

## 2. Method

The first part of this section is dedicated to the tools used for simulating an exhaustive data set for EC and US testing. Afterwards, the ANNs architecture implemented for the data fusion will be introduced. Eventually, the experimental set-up for EC and US measurements will be presented.

### 2.1. Simulated Data Base for EC and US Testing in the Presence of Side Drill Holes (SDHs)

All simulations in this section have been adapted to a homogeneous and isotropic material such as Aluminum block, containing different varieties in diameters and depths of SDHs. This type of material and the flaws shape have been chosen to offer an easy experimental validation. Moreover, the simplified flaw geometries led to calculation time reduction. In the next paragraph, we will report the main aspects of simulating EC testing, using a finite element approach.

#### 2.1.1. Finite Elements EC Simulations

In this paragraph, signals from an Aluminum blocks containing SDHs are obtained via the electric vector potential and magnetic scalar potential $\vec{T}-\Omega$ formulation adapted to multi-connected geometries [32]. This formulation allows us to reduce simulation time by computing electric vector potential only in the conductive domain. The coil supply is sinusoidal, which allows the study in the harmonic regime. In this work, the frequency of the excitation sinusoidal signal has been set to 1 kHz. The magnetic field $\vec{H}$ is expressed through a magnetic source field $\vec{H_{S}}$ and an electric potential vector $\vec{T}$ following (Equation 1) in the conductive domain and () outside. $\vec{H}$ and $\vec{T}$ are linked by () where Ω is the magnetic scalar potential. To obtain a unique solution and because magnetic potential varies from zero to infinity, Ω=0 is imposed on the surface of an air box surrounding the simulated domain. Moreover, $\vec{T}$ is set to $\vec{0}$ on the conductive domain surface. An implicit gauge condition through an iterative resolution algorithm allowed us to obtain convergence and uniqueness of the solution. The electrical conductivity and magnetic relative permeability are defined as σ=37.7 MS·m${}^{-1}$ and $\mu_{r}=1$. These calculations allowed us to compute the induced currents $\vec{J}$, the electric field $\vec{E}$, the magnetic induction $\vec{B}$ and the magnetic field $\vec{H}$. The active and reactive impedance *R* and *X* of the simulated probe are derived from the active dissipated power in the conductive domain $D_{c}$ and the reactive dissipated power in the simulated domain *D* following () and (). This allowed us to compute the impedance variation of the inductor due to the presence of the flaw. To avoid the meshing of the coil, the reaction of the probe to the electromagnetic field generated by EC has not been taken into account during the simulation procedure.

A data set of 6400 different cases is built varying the parameters of cylindrical flaws from a radius of 0.1 to 8 mm and from a depth of 0.1 to 8 mm, depth being the distance between the surface and the top of each flaw. A nonlinear least-squares optimization [33] between simulations and experimental signals presented in Section 2.3 is implemented to estimate the probe parameters, leading to an EC coil composed by a 2.25 mm height and 1400 spires with inner and outer radius of 2.22 and 2.48 mm. Air gap was estimated with the value of 0.18 mm. The skin depth δ, defined by () [12], is equal to 2.59 mm, consequently no SDH is expected to be detected after 7 mm depth, corresponding to 3δ. Eventually, the probe response is simulated every millimeter along the inspection direction.
(1)$$\begin{array}{cc} \vec{J_{ind}} & =\vec{\nabla }\times \vec{T}, \end{array}$$
(2)$$\begin{array}{cc} \vec{T} & =\vec{0}, \end{array}$$
(3)$$\begin{array}{cc} \vec{H} & =\vec{H_{S}}+\vec{T}-\vec{\mathrm{grad}}\left(\Omega \right), \end{array}$$
(4)$$\begin{array}{cc} RI_{source}^{2} & =\int_{D_{c}}\vec{E}\cdot \vec{J}dD_{c}=\int_{D_{c}}\sigma^{-1}\vec{J}\cdot \vec{J}dD_{c}, \end{array}$$
(5)$$\begin{array}{cc} jXI_{source}^{2} & =\int_{D}\frac{\partial \vec{B}}{\partial t}\cdot \vec{H}dD=j\omega \int_{D}\mu^{-1}\vec{B}\cdot \vec{B}dD, \end{array}$$
(6)$$\begin{array}{cc} \delta & =\frac{1}{\sqrt{\pi \mu \sigma f}}. \end{array}$$

#### 2.1.2. Ultrasonic (US) Testing Simulations

In the case of a homogeneous and isotropic medium with limited attenuation, the propagation of elastic waves is given by the differential Equations (Equation 7)–() [8], where $\vec{u}$ is the velocity of the medium particles, *p* the sound pressure, ρ the density of the material and *c* the propagation velocity of the wave. These equations are solved with a pseudo-spectral method [34], by means of the K-WAVE toolbox [35]. Even in this case, simulations have been performed in an Aluminum block, and wave fronts propagation have been observed in the presence of a wide variety of SDHs. The density of test material has been set to ρ=2700 kg·m${}^{-3}$ and the longitudinal wave velocity was previously estimated in v=6542 m·s${}^{-1}$. The spatial resolution of the simulated grid was fixed to ensure the accuracy of the results. The simulated US transducer presented a beam diameter of 6.35 mm and sent out a burst composed of three cycles of sinusoidal shape at the frequency of 5 MHz. Inspection was performed in pulse-echo mode every 1 mm, along the block surface. The presence of transverse waves and longitudinal wave attenuation has not been taken into account because of their negligible effects here.

The flaw profiles were then extracted by estimating the Time of Flight at each sensor position, by looking at the maximum envelope of each Amplitude Scan (A-scan) between the emission and the background echo. The amplitude corresponding to detected Time of Flight was also recorded. As mentioned in the Introduction, one of the limitations of US inspections is the presence of the near field *l* defined by () [8], with *D* the transducer diameter and λ the wavelength. In our case, we could theoretically expect l≈7.7 mm but in the presented simulations this value is prone to be overestimated because the calculations were performed taking into account only two spatial dimensions. To conclude this subsection, a simulated EC and US data base has been generated, to train an ANNs in a data fusion configuration, as presented in the next paragraph.
(7)$$\begin{array}{cc} \frac{\partial \vec{u}}{\partial t} & =-\frac{1}{\rho }\nabla p, \end{array}$$
(8)$$\begin{array}{cc} \frac{\partial p}{\partial t} & =-\rho \nabla \cdot \vec{u}, \end{array}$$
(9)$$\begin{array}{cc} p & =c^{2}\rho , \end{array}$$
(10)$$\begin{array}{cc} l & =\frac{D^{2}}{4\lambda }. \end{array}$$

### 2.2. Artificial Neural Network for Data Fusion

ANNs are employed in this work with the aim to implement a data fusion technique able to estimate flaws parameters using simultaneously EC and US signals. Generally speaking, an ANNs is composed of simple computing units, interconnected by means of coefficients called weights, and able to perform predictions in regression and/or classification contexts, after being trained on an exhaustive data set. The ANNs implemented in this study has been trained using four inputs. The first two are represented by the simulated real and imaginary parts of the complex impedances and are related to the EC testing. The latter ones correspond to the Time of flight and echoes amplitudes derived during the US simulations. The ANNs inputs were then connected to one hidden layer of 20 neurons. This value, being set empirically, assures accurate estimations without risking over-fitting. In our case, deeper architectures with more than one hidden layer and more neurons in hidden layers have been tried without improving the ANN parameters estimations, and even deteriorating them due to over-fitting. The more neurons the more parameters have be optimized, so the more training samples are needed. It is a good practice with ANNs to keep the architecture as small as possible while reaching enough accuracy, as bigger ANNs tend to over-fit. The final layer is composed of two outputs neurons representing the radius and depths estimations as depicted in Figure 1. The activation functions $f_{v}$ and $f_{w}$ reported in (Equation 11) were chosen to form a universal regression algorithm [36].

The ANNs training consisted of minimizing the chosen error function between the network outputs $S_{W}$ and the desired ones *T* over the training samples [37]. To do so, the weighted sum $S_{v}\left(k\right)$ or $S_{w}\left(k\right)$ of each neuron inputs in () and () was calculated for each unit. Successively, an optimization procedure based on the Levenberg-Marquardt back-propagation algorithm [38] has been implemented. Eventually, to better evaluate the estimation errors, simulated data were split into training data set (50%), validation data set (25%) and test data set (25%). After the training procedure, the ANNs has been employed to perform predictions on real signals. The experimental set-up for EC and US measurements is presented in the next subsection.
(11)$$\begin{array}{cc} f_{v}\left(x\right) & =\frac{2}{1+e^{-2x}}-1\;\mathrm{and}\;f_{w}\left(x\right)=ax+b, \end{array}$$
(12)$$\begin{array}{cc} S_{v}\left(k\right) & =f_{v}\left(\sum_{n=1}^{N_{E}}v_{nk}\times E\left(n\right)\right), \end{array}$$
(13)$$\begin{array}{cc} S_{w}\left(k\right) & =f_{w}\left(\sum_{m=1}^{N}w_{mk}\times S_{v}\left(m\right)\right). \end{array}$$

### 2.3. Experimental Set-Up

EC and US measurements are performed on an Aluminum block containing three cylindrical defects with radius of 2.5 mm and located at depths of 1, 3 and 7.5 mm as shown in Figure 2. These three defects were chosen to highlight the advantages of a data fusion technique, as the first and third defect are detectable only by EC and US respectively, and the second one is situated in the blind region of both techniques. Due to the distance between the different defects, the presence of parasite interactions in the recorded signals has been considered negligible.

#### 2.3.1. Experimental Set-Up for EC Measurements

The experimental set-up used for EC testing is composed by a NORTEC 600 connected to a low-frequency Olympus probe, a data acquisition device (PicoScope 5244B—bandwidth 200 MHz, 16 bit). The EC probe is placed in contact with the test material, perpendicular to its surface, and it can move along the inspection direction thank to a step motor. For every measured point, the NORTEC 600 induced an electromagnetic field at the frequency of 1 kHz through the probe and measured the resulting impedance. Active and reactive impedances were then recorded after an averaging process (number of averages = 100) and sent to the PC station via USB connection. The entire experimental set-up is depicted in Figure 3.

For illustrative purposes, the corresponding measured and simulated signals are compared in the Figure 4. One could notice that the signals recorded for the first flaw located at a depth of 1 mm depicted in Figure 4a are close to the simulated ones. One the other hand, the signals from the second flaw at a depth of 3 mm in Figure 4b show a larger deviation, especially for the reactive impedance. This difference is because the simulations did not take into account the reaction of the probe to the field generated by EC, as already mentioned in Section 2.1. Eventually, due to the skin depth δ, the third defect presented in Figure 4c and located at a depth of 7.5 mm did not register a significant reaction to the excitation signal.

#### 2.3.2. Experimental Set-Up for US Measurements

The experimental device for US measurements is composed by a pulse-echo acquisition system, equipped with a 6.35 mm diameter US probe (central frequency of 5 MHz), a data acquisition device (PicoScope 5244B—bandwidth 200 MHz, 16 bit), a step motor and a PC station, as depicted in the schematic diagram of Figure 5. The US probes, connected to the Picoscope Arbitrary Wave Form Generator, was placed in contact with the surface material via a coupling gel, and it can move with a spatial step of 1 mm along the inspection surface thanks to a step motor. For every measurement point, the US probe sent out a burst of three sinusoidal cycles at the frequency of 5 MHz. Eventually, the echoes generated by the presence of the flaws were recorded after an averaging process (number of averages = 100) and sent to the PC station via USB connection.

A comparison between the simulated Time of Flight in function of the position and the measured ones is given in Figure 6. As expected, the first defect located 1 mm depth was completely hidden because of the blind zone (Figure 6a). The echo of the defect located at 3 mm depth is partly in the blind zone, its Time of Flight is still measurable thanks to the coupling at the interface transducer-material (Figure 6b). Eventually, as shown in Figure 6c, the Time of Flight recorded for the third defect, located at 7.5 mm, agrees with the simulations of Section 2.1. In the next section we will present the obtained estimations of the flaw parameters, considering both simulated and real measurements.

## 3. Results and Discussion

### 3.1. Preliminary Results Using Simulated Data

This paragraph presents the results obtained using EC and US data using the ANNs architecture reported in Section 2.2. The input data is represented by the simulated complex impedances of the EC, together with the Time of Flight and amplitudes of the US. The obtained ANNs outputs represent the estimated radius and depth of the simulated flaws. Eventually, cross-validation is performed to observe the mean and standard deviation of the Mean Square Errors (MSEs) of each set.

The results are summarized in Table 1. The 10 cross-validations show an average MSE of 0.038 mm${}^{2}$ for training and validation data sets, with standard deviations of 0.002 and 0.005 mm${}^{2}$. The test data set presents an average MSE of 0.040 mm${}^{2}$ and a standard deviation of 0.007 mm${}^{2}$. As example, the Figure 7 compares the estimations of flaws radii and depths with the expected values for one cross-validation test set, corresponding to an average MSE of 0.034 mm${}^{2}$ and a Root Mean Square Error (RMSE) of 0.18 mm. An additional visualization of these preliminary results could be given by Figure 8a,b, which compare estimated and real radii and depths corresponding to an Mean Absolute Error (MAE) for estimated radii of 0.20 mm and 0.052 mm for the estimated depths.

To better monitor the accuracy of the flaw parameters estimations, the MAEs for estimated radii and depths were averaged by zones, as depicted in Figure 9a,b. One could notice that the MAE for radius estimations varies from 0.038 to 0.56 mm and for depths from 0.01 to 0.16 mm. In example, flaw parameters estimations of 2 mm radius and 1.5 mm depth, presented a MAE of 0.097 mm and 0.053 mm, respectively.

### 3.2. Final Results Using Experimental Data

To validate the proposed data fusion approach, the EC and US data measured on the Aluminum test sample containing the three SDHs (see Section 2.3), are used to test the ANNs trained with simulated data from Section 2.1. As mentioned in Section 2.3, these flaws presented a radius of 2.5 mm and were respectively located at depths of 1, 3 and 7.5 mm (see Figure 2). The radius and depth estimations are reported in Table 2, Table 3 and Table 4. These Tables report that the first SDH is estimated with a radius of 2.51 and a depth of 1.08 mm (Table 2). On the other hand, the second one is estimated with a radius of 2.56 and a depth of 2.55 mm (Table 3). Eventually, the third SDH, is estimated with a radius of 1.91 and a depth of 7.59 mm (Table 4). It is important to point out that an accurate estimation for the second flaw could not have been achieved only using US and EC separately, this second SDH being located near the blind zone of each method.

## 4. Conclusions

In this paper, it has been shown that use of ANNs allowed us to implement a data fusion algorithm for NDT applications using simultaneously EC and US data, with the aim to exploit the complementary advantages of these two different techniques.

At first, the developed method was tested with simulated signals, estimating radii and depths of SDHs in an Aluminum block presenting a wide variety of radius and depth. In addition, a statistical study of the ANNs performance has been effectuated, providing a confidence index for each estimation. Eventually, experimental measurements were performed to validate the entire procedure. The experimental results confirmed that proposed data fusion algorithm is able to properly estimate SDHs depths and radii, even for flaws located in the EC and US blind zone. Despite the simplicity of the method presented in this paper in terms of ANNs architecture, test material and the flaw geometries, we believe that the approach presented here could represent a promising tool for future employments of data fusion algorithms in NDT. Further improvements could be tied to more accurate numerical models and multi-layers ANNs, with the aim to investigate more complex materials presenting different flaw geometries. This would reduce estimations errors tied to different levels of uncertainty and inaccuracy in the data set. The ultimate goal of this study could be the design of an NDT smart sensor, able to provide accurate estimations of flaws eventually present in the structure, combing EC and US inspections.

## Figures and Tables

**Figure 1 sensors-21-02598-f001:**
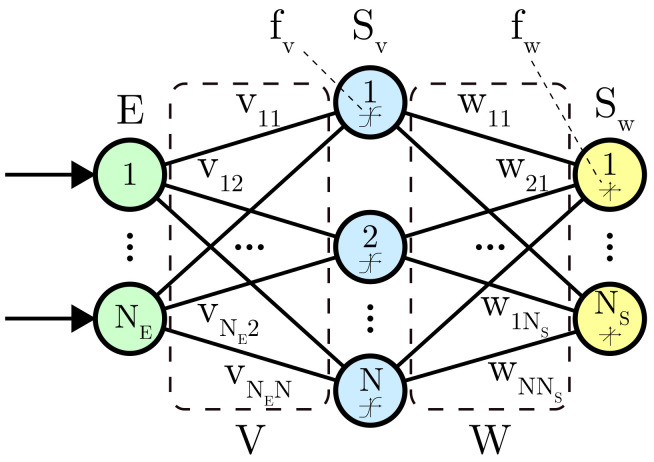
The ANNs architecture implemented this work, composed by one hidden layer of 20 neurons.

**Figure 2 sensors-21-02598-f002:**
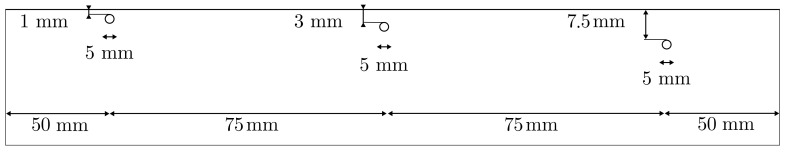
Schematic representation of the test sample presenting three SDHs of 2.5 mm radius, located at depths of 1, 3 and 7.5 mm depth.

**Figure 3 sensors-21-02598-f003:**
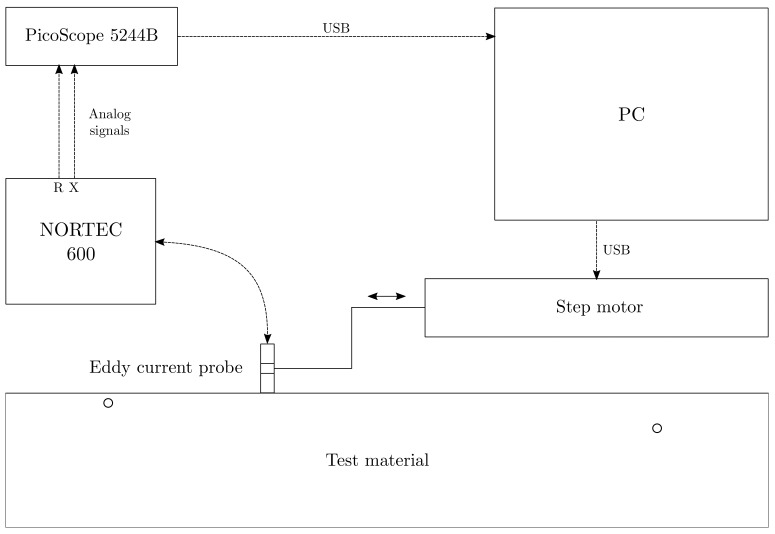
Experimental set-up for EC measurements, composed by a NORTEC 600 connected to a low-frequency Olympus probe and a data acquisition device (PicoScope 5244B—bandwidth 200 MHz, 16 bit).

**Figure 4 sensors-21-02598-f004:**
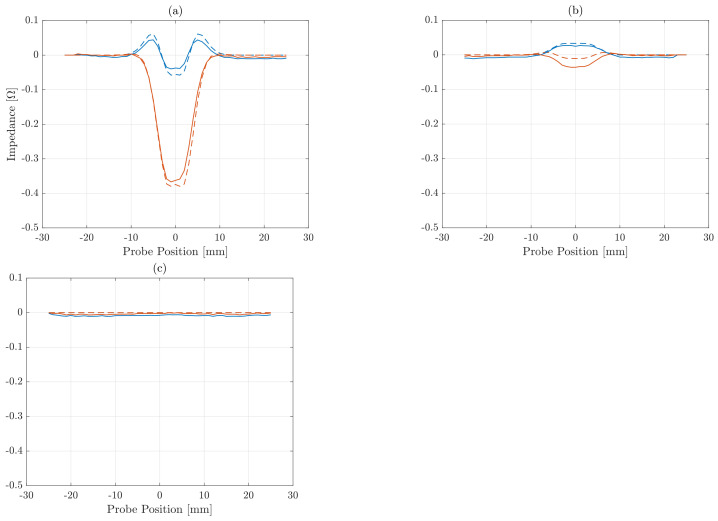
Active impedances *R* (red) and reactive impedances *X* (blue) obtained experimentally (−) and by numerical simulations (−−) for three SDHs of 2.5 mm radius and located at 1 (**a**), 3 (**b**) and 7.5 mm (**c**) depth.

**Figure 5 sensors-21-02598-f005:**
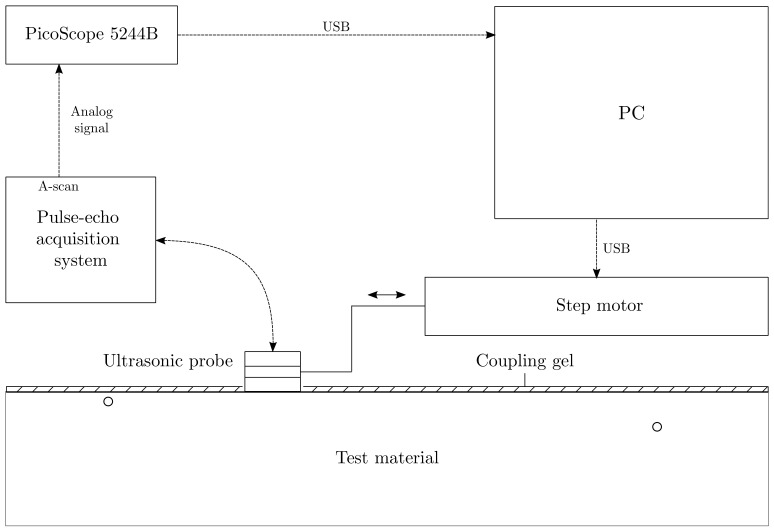
Experimental set-up for US measurements, composed by a pulse-echo acquisition system, equipped with a 6.35 mm diameter US probe (central frequency of 5 MHz) and a data acquisition device (PicoScope 5244B—bandwidth 200 MHz, 16 bit).

**Figure 6 sensors-21-02598-f006:**
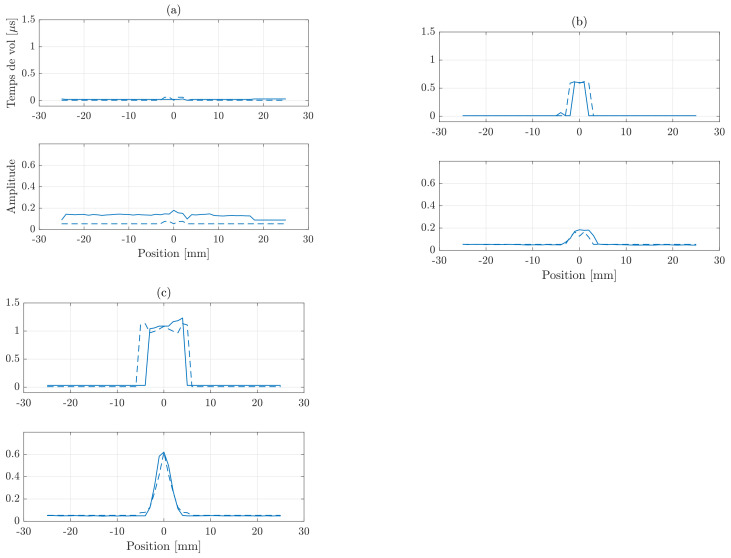
Time of Flight and echoes amplitudes obtained experimentally (−) and by numerical simulations (−−) for three SDHs of 2.5 mm radius and located at 1 (**a**), 3 (**b**) and 7.5 mm (**c**) depth.

**Figure 7 sensors-21-02598-f007:**
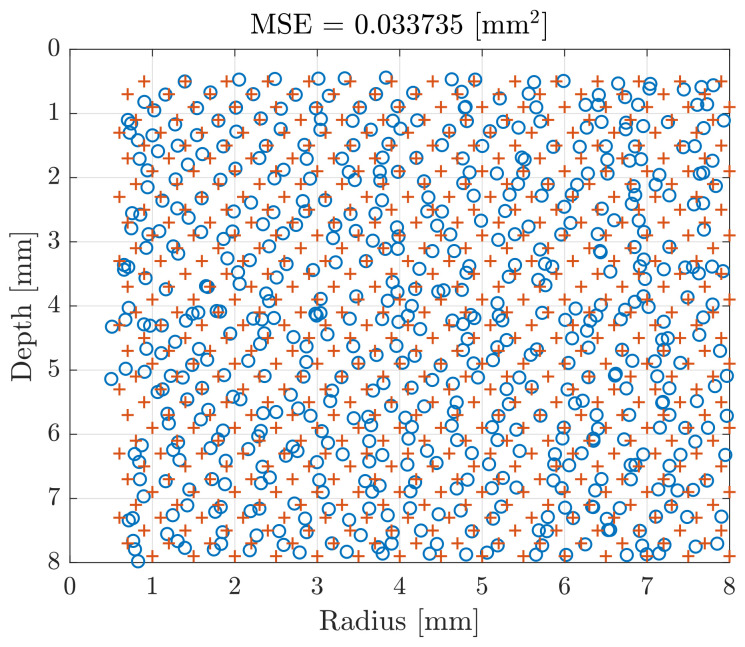
Estimated radii and depths (circles) compared with real parameters (crosses) for the test data using the data fusion EC and US.

**Figure 8 sensors-21-02598-f008:**
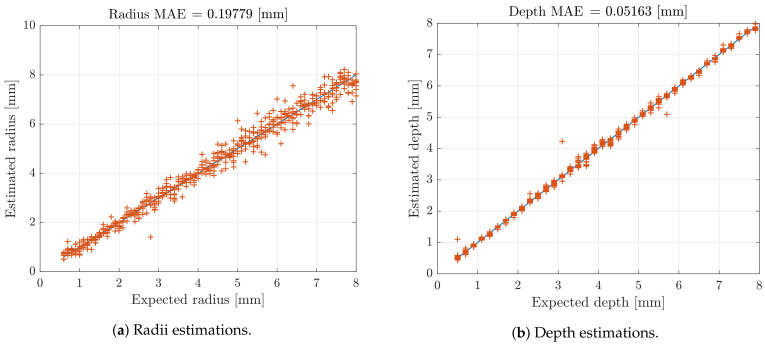
Estimated and expected radii and depths for the test data set.

**Figure 9 sensors-21-02598-f009:**
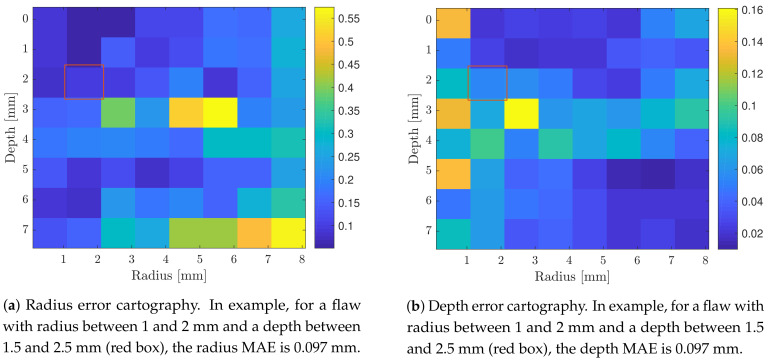
Cartography of the MAEs for estimated radii and depths in function of the real flaw parameters.

**Table 1 sensors-21-02598-t001:** Mean values and standard deviations of the MSEs obtained by cross-validation for the data fusion between EC and US.

	MSE
Training data set	0.038 ± 0.002 mm${}^{2}$
Validation data set	0.038 ± 0.005 mm${}^{2}$
Test data set	0.040 ± 0.007 mm${}^{2}$

**Table 2 sensors-21-02598-t002:** ANNs estimations of radii and depths (mm) obtained for the first measured flaw.

Flaw 1	Radius	Depth
True values	2.5	1
Data fusion estimation	2.52 ± 0.18	1.08 ± 0.07

**Table 3 sensors-21-02598-t003:** ANNs estimations of radii and depths (mm) obtained for the second measured flaw.

Flaw 2	Radius	Depth
True values	2.5	3
Data fusion estimation	2.56 ± 0.27	2.55 ± 0.29

**Table 4 sensors-21-02598-t004:** ANNs estimations of radii and depths (mm) obtained for the third measured flaw.

Flaw 3	Radius	Depth
True values	2.5	7.5
Data fusion estimation	1.91 ± 0.18	7.59 ± 0.10

## Data Availability

Not applicable.

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
