# Peer review of "A Data Fusion Method for Non-Destructive Testing by Means of Artificial Neural Networks"

_sensors, 2021, doi:10.3390/s21082598_

Round 1
Reviewer 1 Report
- The abstract is concise and contains all the qualitative information including the most interesting results. However, it is advisable to include at least one sentence in which the main quantitative results are summarized.
- In Section 2.1.1. some formulas appear in the text. If the Authors deem it, they can separate the formulas from the text in order to make the Section more readable.
- Please make the captions self-explanatory.
- The proposed method is certainly interesting and offers food for thought for future developments. However, what if the data is affected by uncertainty and / or inaccuracy? Is the proposed methodology suitable for manipulating such data? Obviously the answer to this question cannot be given in this paper but it could be a suggestion for future work. However, I advise the authors to include at least one sentence in the text to highlight this possibility by highlighting that neuro-fuzzy techniques could be useful. I also recommend that you include the following relevant work in the bibliography:
doi: 10.1155/2014/201243
Reviewer 2 Report
This paper presented a new data fusion method for NDT using ANN. While this paper provides novel methods, the reviewer believes the issues below need to be addressed to be published in this journal.
- Is this the first attempt to combine US and EC data and use Machine Learning to estimate the damage? If not, please add more literatures.
- Is the proposed ANN composed of 20 hidden layers or a single hidden layer with 20 nodes? In this former case, people generally call the network as DNN instead of ANN. While there is no strict definition for Deep Learning, most people consider an ANN network with more than 2 hidden layer as a Deep Neural Network. I highly recommend the authors to revise the manuscript accordingly, and introduce the terminology for DNN.
- Are all hidden layers fully connected? Please describe more on the network configuration. The reviewer suggest to add a figure describing the network configuration.
- Why did the authors select 20 neurons? Please discuss more about the numbers of hidden layers and hidden units. (or show the result with different values)
Reviewer 3 Report
The authors did not provide satisfied improvements at the re-submission process. Still the following weaknessess were unanswered:
- Abstract is not well written. It should rather presents what is the aim of the article and what are the main findings. In its current form almost 90% of the abstract shows the motivation. In my opinion the abstract should be re-written to presents what is the aim of the article and what are the main findings,
- The novelty of the study is limited as it is well-known that ANN should be used complementary and ANNs are useful for this purpose,
- The article does not have any literature survey. Unfortunately there are too many citation pockets (e.g. [4-7]) without deep analysis of each single reference and without justification of the novelty of the study presented by authors,
- The data for ANN numerical modelling were simulated (mostly by FEM). Thus, in my opinion the significance of the study is limited,
- The conclusions are too short and are not supported by the data,
- The obtained model was not validated. Thus, in my opinion its significance is limited.
Round 2
Reviewer 3 Report
The manuscript is acceptable.
This manuscript is a resubmission of an earlier submission. The following is a list of the peer review reports and author responses from that submission.
Round 1
Reviewer 1 Report
The article is devoted to a problem of complementary use of ANNs as data fusion method for non-destructive testing. Even if the topic of the article may be interesting, I see some serious limitations that decrease the publication potential of the manuscript. Therefore, I suggest to reject it at this stage. Below are the main reasons of this recommendation:
- Abstract is not well written. It should rather presents what is the aim of the article and what are the main findings. In its current form almost 90% of the abstract shows the motivation. In my opinion the abstract should be re-written to presents what is the aim of the article and what are the main findings,
- The novelty of the study is limited as it is well-known that ANN should be used complementary and ANNs are useful for this purpose,
- The article does not have any literature survey. Unfortunately there are too many citation pockets (e.g. [15-26]) without deep analysis of each single reference and without justification of the novelty of the study presented by authors,
- The data for ANN numerical modelling were simulated (mostly by FEM). Thus, in my opinion the significance of the study is limited,
- The conclusions are too short and are not supported by the data,
- The obtained model was not validated. Thus, in my opinion its significance is limited.
Reviewer 2 Report
The authors have performed a paper “A data fusion method for Non Destructive Testing by means of Artificial Neural Networks”. The paper presents a combined Eddie current and ultrasonic non-destructive method for detection of defects on the surface of materials applied in aircrafts. They have used an aluminum sample with known holes to prove their method.
I believe that this paper is suitable for publication except some minor comments.
- Authors claim that “this technique presents a blind zone for the acoustic field, located near the sensor”. I would like to have a specific reference discussing the flaw.
- The reference list 15-26 accompanies only one phrase “Some works already offered data fusion techniques for NDT”. Some of them are in French, others are conference proceedings and even PhD works. I believe that 3-4 good books or reviews would replace all these references, if they do exist.
3.The same concerns references 27-37. 3-4 fundamental works on the topic or reviews would replace these 10 references.
- Some references are in French. This should be indicated in the reference list as (in French) at the end of each such reference.
- Minor typos and mistakes
Abstract
“technique such Eddy Current (EC) can be used only” should be “technique such as Eddy Current (EC) can be used only”
Everywhere:
“allows to do smth” should be “allow us to do smth”
Reviewer 3 Report
This paper introduced a data fusion method for NDT by using ANN. While this paper introduced interesting topic, the reviewer believes the issue below needs to be resolved to be published in this journal.
- Introduction should be re-written. More research background is needed. Why is this research important? How can this research impact the society. More literature review is needed. What are the cutting edge technology for NDT? How is ANN or DNN being applied in this area?
- What is the originality of this paper? Is this the first attempt to apply ANN to EC and US?
- The authors mentioned that total a hidden layer of 20 neurons was used. Is this a Multi Layer Perceptron (MLP)? If so, why didn't you try multiple hidden layers?
- More explanation is needed for section 3.2 (experimental result).
- The paper is not well-organized. Only two paragraphs for introduction, and one for conclusion. For section 3.2, one paragraph is consist of single sentence. The structure of the paper should be improved.